# The gut microbiome in atherosclerotic cardiovascular disease

Zhuye Jie[1,2,3], Huihua Xia[1,2], Shi-Long Zhong[4,5], Qiang Feng[1,2,6,7,17], Shenghui Li[1], Suisha Liang[1,2], Huanzi Zhong [1,2,3,7], Zhipeng Liu[1,8], Yuan Gao[1,2], Hui Zhao[1], Dongya Zhang[1], Zheng Su[1], Zhiwei Fang[1], Zhou Lan[1], Junhua Li [1,2,3,9], Liang Xiao[1,2,6], Jun Li[1], Ruijun Li[10], Xiaoping Li[1,2], Fei Li[1,2,8], Huahui Ren[1], Yan Huang[1], Yangqing Peng[1,18], Guanglei Li[1], Bo Wen [1,2], Bo Dong[1], Ji-Yan Chen[4], Qing-Shan Geng[4], Zhi-Wei Zhang[4], Huanming Yang[1,2,11], Jian Wang[1,2,11], Jun Wang[1,12,19], Xuan Zhang [13], Lise Madsen [1,2,7,14], Susanne Brix [15], Guang Ning[16], Xun Xu[1,2], Xin Liu [1,2], Yong Hou [1,2], Huijue Jia [1,2,3,12], Kunlun He[10] & Karsten Kristiansen[1,2,7]

The gut microbiota has been linked to cardiovascular diseases. However, the composition and functional capacity of the gut microbiome in relation to cardiovascular diseases have not been systematically examined. Here, we perform a metagenome-wide association study on stools from 218 individuals with atherosclerotic cardiovascular disease (ACVD) and 187 healthy controls. The ACVD gut microbiome deviates from the healthy status by increased abundance of *Enterobacteriaceae* and *Streptococcus* spp. and, functionally, in the potential for metabolism or transport of several molecules important for cardiovascular health. Although drug treatment represents a confounding factor, ACVD status, and not current drug use, is the major distinguishing feature in this cohort. We identify common themes by comparison with gut microbiome data associated with other cardiometabolic diseases (obesity and type 2 diabetes), with liver cirrhosis, and rheumatoid arthritis. Our data represent a comprehensive resource for further investigations on the role of the gut microbiome in promoting or preventing ACVD as well as other related diseases.

[1] BGI-Shenzhen, Shenzhen 518083, China. [2] China National Genebank, Shenzhen 518120, China. [3] Shenzhen Key Laboratory of Human Commensal Microorganisms and Health Research, BGI-Shenzhen, Shenzhen 518083, China. [4] Guangdong Provincial Key Laboratory of Coronary Heart Disease Prevention, Guangdong Cardiovascular Institute, Guangzhou 510080, China. [5] Medical Research Center of Guangdong General Hospital, Guangdong Academy of Medical Sciences, Guangzhou 510080, China. [6] Shenzhen Engineering Laboratory of Detection and Intervention of Human Intestinal Microbiome, Shenzhen 518083, China. [7] Department of Biology, Laboratory of Genomics and Molecular Biomedicine, University of Copenhagen, Universitetsparken 13, 2100 Copenhagen, Denmark. [8] BGI Education Center, University of Chinese Academy of Sciences, Shenzhen 518083, China. [9] School of Bioscience and Biotechnology, South China University of Technology, Guangzhou 510006, China. [10] Beijing Key Laboratory for Precision Medicine of Chronic Heart Failure, Chinese PLA General Hospital, Beijing 100853, China. [11] James D. Watson Institute of Genome Sciences, Hangzhou 310000, China. [12] Macau University of Science and Technology, Macau 999078, China. [13] Department of Rheumatology and Clinical Immunology, Peking Union Medical College Hospital, Chinese Academy of Medical Sciences and Peking Union Medical College, Beijing 100730, China. [14] National Institute of Nutrition and Seafood Research, (NIFES), Postboks 2029, Nordnes, N-5817 Bergen, Norway. [15] Department of Biotechnology and Biomedicine, Technical University of Denmark (DTU), 2800 Kongens Lyngby, Denmark. [16] Department of Endocrinology and Metabolism, State Key Laboratory of Medical Genomes, National Clinical Research Center for Metabolic Diseases, Shanghai Clinical Center for Endocrine and Metabolic Diseases, Shanghai Institute of Endocrine and Metabolic Diseases, Ruijin Hospital, Shanghai Jiao Tong University School of Medicine, Shanghai 200025, China. [17] Present address: Department of Human Microbiome, School of Stomatology, Shandong University, Shandong Provincial Key Laboratory of Oral Tissue Regeneration, Jinan 250012, China. [18] Present address: Center for Genome Sciences & Systems Biology, Washington University School of Medicine, St. Louis, MO 63110, USA. [19] Present address: iCarbonX, Shenzhen 518053, China. Zhuye Jie, Huihua Xia, Shi-Long Zhong and Qiang Feng contributed equally to this work. Correspondence and requests for materials should be addressed to H.J. (email: jiahuijue@genomics.cn) or to K.H. (email: hekl301@aliyun.com) or to K.K. (email: kk@bio.ku.dk)

Cardiovascular and metabolic diseases, collectively referred to as cardiometabolic diseases (CMDs), are associated with high morbidity and mortality as well as with considerable and increasing health-care costs[1]. The gut microbiome has emerged as a central factor affecting human health and disease[2, 3], and CMDs are no exception. Pioneering metagenomic shotgun-sequencing studies have enabled characterization of the gut microbiome in type 2 diabetic and obese subjects[3–9], and furthered our understanding of the functional interplay between the gut microbiota and host physiology. By contrast, only a very small number of samples from patients with cardiovascular diseases have been analyzed[10]. Previous studies have shown that the gut microbiota metabolizes choline, phosphatidylcholine, and L-carnitine to produce trimethylamine (TMA), which is oxidized in the liver into the proatherogenic metabolite, trimethylamine-N-oxide (TMAO)[11–13]. Inhibition of gut microbiota-dependent TMAO production has been shown as a promising strategy for the treatment of atherosclerosis[14]. Bacterial DNA has also been detected in atherosclerotic plaques[15–17]. However, the lack of a large cohort for metagenomics characterization of this major group of CMD has impeded further investigations on the role played by the microbiome.

Here, we sequenced stool samples, representative of the gut microbiome, from 218 individuals with atherosclerotic cardiovascular disease (ACVD) and 187 healthy controls, and performed a metagenome-wide association study (MWAS) identifying strains (metagenomic linkage groups (MLGs))[3, 4] and functional modules associated with ACVD. Integrative analyses of an additional 845 samples from other disease cohorts revealed common alterations suggestive of a less fermentative and more inflammatory gut environment in ACVD, type 2 diabetes (T2D), obesity and liver cirrhosis, in contrast to the autoimmune disease rheumatoid arthritis (RA).

## Results

**Composition of the ACVD gut microbiome**. In order to investigate the gut microbiome in ACVD patients, we performed metagenomic shotgun sequencing on a total of 405 fecal samples from 218 individuals with ACVD (defined as ≥50% stenosis in one or more vessels) and 187 healthy controls (Supplementary Data 1). After removal of low-quality reads and human DNA reads, 2.2 Tb of high-quality sequencing reads (on average 55.2 million reads per sample) were aligned to a comprehensive reference gut microbiome gene catalog comprising 9.9 million genes[18], which allowed on average $80.0 \pm 3.5\%$ of the reads in each sample to be mapped (Supplementary Data 2), consistent with saturation of the gene-coding regions[4, 18].

The ACVD and control samples were significantly different in multivariate analyses. ACVD status showed a $P$-value $<10^{-6}$ in permutational multivariate analysis of variance (PERMANOVA), with or without adjustment for medication (Supplementary Table 1). The ACVD and control samples also showed separation in PCA (principal component analysis) and dbRDA (distance-based redundancy analysis) plots (Fig. 1), which was corroborated by abundance differences between the two groups in major genera of the gut microbiome, such as a relative reduction in Bacteroides and Prevotella, and enrichment in Streptococcus and Escherichia in ACVD (Fig. 1b).

Despite the reduction in major genera and possible overgrowth of rare genera in ACVD, we observed no significant difference between the ACVD and control samples in either gene richness or diversity (Supplementary Fig. 1).

**Microbial strains associated with ACVD**. To identify the microbial species or strains associated with ACVD, we clustered the 9.9 million genes into 2982 MLGs (containing >100 genes) according to co-variations of their abundances among the 405 samples, based on the idea that genes from the same microbial genome are physically linked[3, 4, 18, 19]. This is so far the largest cohort and the largest reference gene catalog for co-abundance-based binning. A total of 536 of the MLGs differed significantly in abundance between ACVD and control samples (Wilcoxon-rank sum test, Benjamin–Hochberg $q$-value <0.05), with 320 of these being more abundant in ACVD samples (Fig. 2, Supplementary Data 3). The 536 differentially enriched MLGs represented on average 56.5% of the relative abundance in all MLGs, confirming the major compositional differences between the ACVD and control samples.

The abundance of Enterobacteriaceae including Escherichia coli, Klebsiella spp., and Enterobacter aerogenes, was higher in ACVD than in control samples ($q$-value < 0.05, Fig. 2, Supplementary Data 3). The relative abundance of bacteria that are often present in the oral cavity, such as Streptococcus spp., Lactobacillus salivarius, Solobacterium moorei, and Atopobium parvulum, was also higher in patients with ACVD than in healthy controls ($q$-value <0.05, Fig. 2, Supplementary Data 3). The abundance of Ruminococcus gnavus, a bacterium previously associated with inflammatory bowel diseases and low gut microbial richness[8, 20–22] was higher in ACVD samples than in control samples ($q$-value <0.05, Fig. 2, Supplementary Data 3). The abundance of Eggerthella lenta, which has been reported to possess enzymes for deactivating the cardiac drug digoxin[4, 23–25], was higher in ACVD ($q$-value <0.05, Fig. 2, Supplementary Data 3). In contrast, butyrate-producing bacteria including Roseburia intestinalis and Faecalibacterium cf. prausnitzii were relatively depleted in the ACVD samples ($q$-value <0.05, Fig. 2, Supplementary Data 3). Consistent with the genera results (Fig. 1), common members of the gut microbiome such as Bacteroides spp., Prevotella copri, and Alistipes shahii were also relatively depleted in ACVD ($q$-value <0.05, Fig. 2, Supplementary Data 3).

Besides abundance differences between ACVD and control samples, the MLGs also showed differences in network structure (Spearman's correlation coefficient (cc) ≥0.3 or ≤−0.3, Fig. 2). Most notably, the ACVD-enriched aerobes Streptococcus spp. showed negative correlations with the ACVD-depleted commensals Bacteroides spp. only in the ACVD samples, and the positive associations in controls between Bacteroides spp., Lachnospiraceae bacterium, and Erysipelotrichaceae bacterium were concomitantly diminished in ACVD. Meanwhile, the Streptococcus spp. clusters displayed more positive correlations with the ACVD-enriched Enterobacteriaceae cluster. The ACVD-enriched cluster of Eggerthella spp., R. gnavus, Clostridium spp., Erysipelotrichaceae bacterium, and Lachnospiraceae bacterium showed more negative associations with the ACVD-depleted butyrate-producing bacteria including Eubacterium eligens, F. prausnitzii, and Clostridiales sp. SS3/4 (Fig. 2). R. gnavus and Lachnospiraceae bacterium also negatively associated with A. shahii. These results demonstrated profound imbalances in the composition and inter-species relationship in the gut microbiome of ACVD patients as compared to healthy controls.

**Links between the gut microbiome and clinical features of ACVD**. To explore the diagnostic value of the gut microbiome composition in relation to ACVD, we constructed a random forest classifier from the 405 ACVD and control samples, with five repeats of fivefold cross-validation (RFCV, Fig. 3). The area under receiver operating curve (AUC) was 0.86 in this ACVD cohort. Among the 47 MLGs selected by the ACVD classifier, the MLGs most important for the classifier belonged to Streptococcus

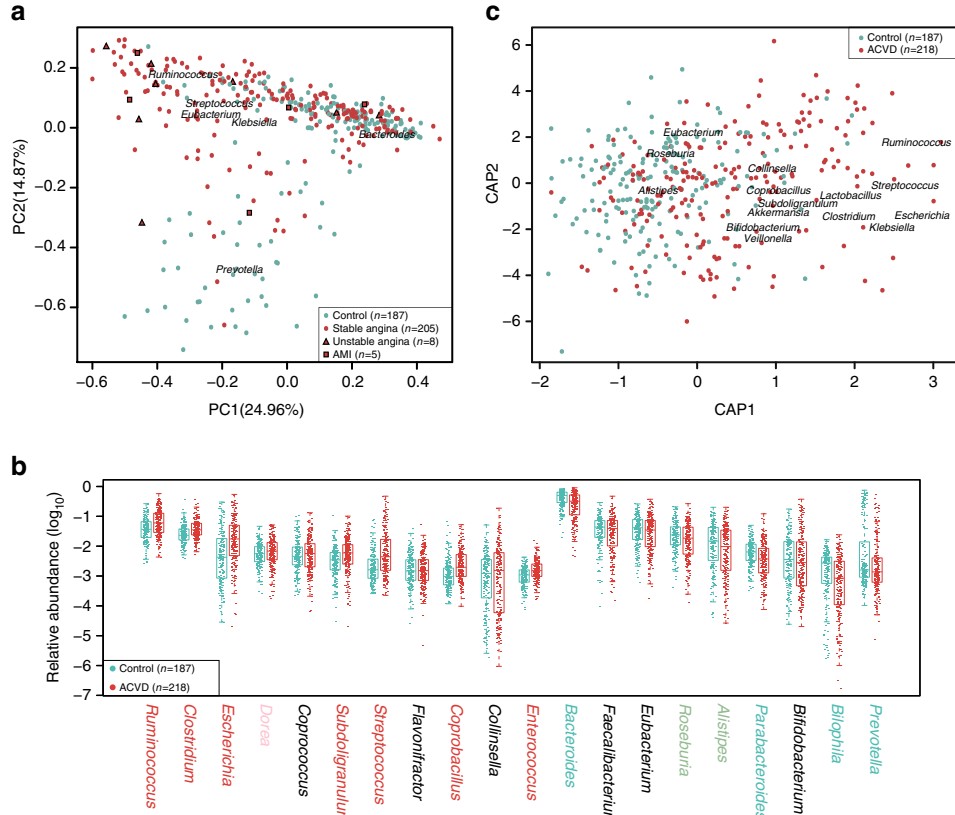

**Fig. 1** Major genera in the ACVD gut microbiome. **a** PCA of genus-level ACVD gut microbiomes. Control samples, n = 187 (*cyan*); ACVD samples, n = 218 (*red*). For the ACVD samples (Supplementary Data 1), 205 were stable angina (*circles*), 8 were unstable angina (*triangles*), and 5 were acute myocardial infarction (AMI) (*squares*). Genera with the largest weights on each principal component are shown. **b** Relative abundances of the top 20 most abundant genera. The genus names were colored according to significant differences between the ACVD and control samples, i.e., *red* or *cyan*, q-value <0.01; *light red* or *green*, q-value <0.05; *black*, q-value ≥0.05, Wilcoxon rank-sum test, controlled for multiple testing. *Boxes* represent the median and interquartile ranges (IQRs) between the first and third quartiles; *whiskers* represent the lowest or highest values within 1.5 times IQR from the first or third quartiles. *Circles* represent all data points. **c** Differentially changed genera in ACVD and controls according to dbRDA based on Bray–Curtis distance. Genera with the largest weights on each principal coordinate are shown. CAP constrained analysis of principal coordinates

*vestibularis*, *E. lenta*, *A. parvulum*, *R. gnavus*, *L. salivarius*, *E. coli*, *Lachnospiraceae*, and *Clostridium nexile* (Supplementary Data 3). While validation of the identified markers would require independent and much larger cohorts, these results demonstrate the presence of ACVD-associated features in the gut microbiome that may be further developed into non-invasive and inexpensive biomarkers. Concomitantly, we noted that the MLG-based classifier had an AUC that was larger than the AUC of 0.63 using TMA lyases only (CutC/D and YeaW/X, Fig. 3), indicating that factors in addition to TMAO are implicated in ACVD. According to further analysis using the reference-genome based method PanPhlAn[26], ACVD-enriched bacteria encoding choline-TMA lyase (CutC) included an unclassified *Erysipelotrichaceae* bacterium, *C. nexile*, and *S. anginosus* (Fig. 2, Supplementary Fig. 2a). ACVD-enriched bacteria encoding both the choline-TMA lyase and the more promiscuous TMA lyase (YeaW/X, carnitine, choline, and betaine) included *E. aerogenes* and *Klebsiella pneumoniae* (Fig. 2, Supplementary Fig. 2a). In addition to TMA lyases, a number of virulence factors[27] in these bacteria might also play a role, such as immunogenic lipoprotein A *IlpA*, and *PhoP*, part of the *PhoQ/PhoP* two-component system that could be induced by host antimicrobial peptides[28] (Supplementary Fig. 2b).

Besides being able to distinguish between individuals with and without ACVD, the fecal MLGs showed associations with a number of clinical indices (Spearman correlation permutational P < 0.05, Spearman's cc ≥0.2 or ≤−0.2, corroborated by RFCV

selections, Fig. 4). *K. oxytoca* that showed increased abundance in ACVD patients, correlated positively with serum levels of aspartate transaminase (AST, a marker for acute myocardial infarction as well as other conditions), α-hydroxybutyrate dehydrogenase (HBDH), and creatine kinase (CKMB). *K. pneumoniae* and *Bifidobacterium dentium* also positively correlated with HBDH. ACVD-enriched bacteria including *Streptococcus* sp. C300, *Streptococcus* sp. oral taxon 071 73H25Ap, *S. salivarius*, *Oribacterium sinus*, and *Clostridium perfringens* positively correlated with diastolic blood pressure or systolic blood pressure (Fig. 4). *F. cf. prausnitzii* that was depleted in ACVD patients correlated negatively with serum levels of uric acid, which has been reported to increase after a diet rich in red meat[29] and decrease after intake of the DASH diet (dietary approaches to stop hypertension)[30]. *Clostridium hathewayi* correlated with the heme catabolism product serum total bilirubin (TBIL).

When KEGG (The Kyoto Encyclopedia of Genes and Genomes)[31] functional modules were selected to construct a mathematical model that predicts clinical indices in the samples (cross-validated group LASSO (least absolute shrinkage and selection operator) with bootstrapping[32], modules that were most important for the clinical indices were not necessarily most important for identifying ACVD (Supplementary Fig. 2c). Yet, the most important module for ApoE (apolipoprotein E), LDL (low-density lipoprotein) cholesterol, and total cholesterol (TC)

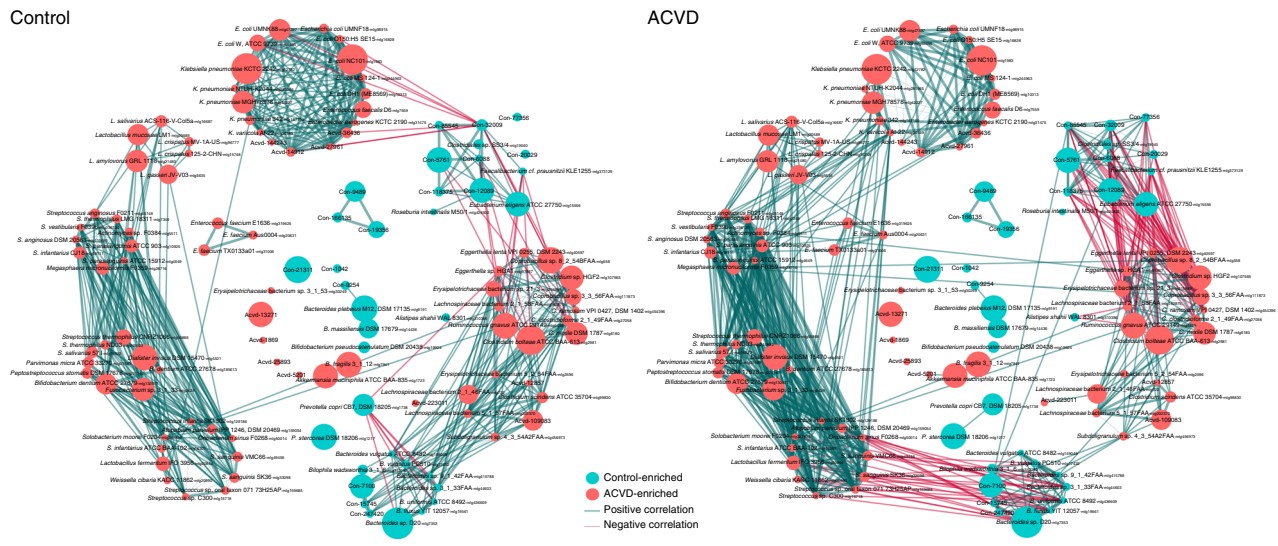

**Fig. 2** Co-abundance network of MLGs differentially enriched in individuals with and without ACVD. *Left*, network in healthy controls (*n* = 187); *right*, network in individuals with ACVD (*n* = 218), arranged in the same order. MLGs (>100 genes) whose relative abundances were significantly different between the groups are shown (*q*-value <0.05, FDR-controlled Wilcoxon rank-sum test). *Red circles*, ACVD-enriched; *cyan circles*, control-enriched. The size of each circle indicates the number of genes in an MLG (100–3723). MLGs not annotated to a known species are shown with their identification number only. Please see Supplementary Data 3 for more information on taxonomic annotations. *Green edges*, positive correlations; *red edges*, negative correlations. The width of the edges decreases with the absolute value of the Spearman's cc: *thick edges*, |cc| > 0.7; *medium*, 0.5 < |cc| < 0.7; *thin*, 0.3 < |cc| < 0.5

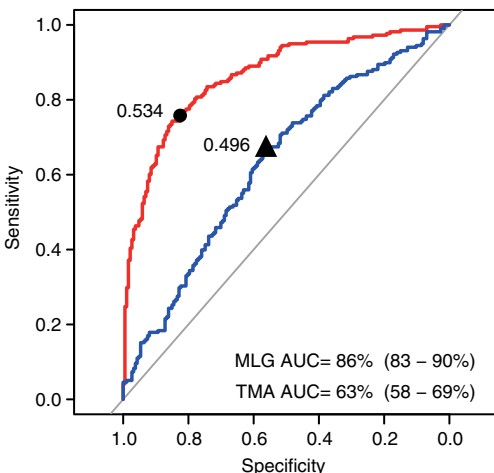

**Fig. 3** Gut microbiome-based identification of ACVD. Receiver operating curve (ROC, *red*) according to cross-validated random forest models on MLGs (fivefold RFCV performed five times) from stool samples of 218 ACVD and 187 healthy individuals. The 47 MLGs selected are shown in Supplementary Fig. 5 and Supplementary Data 3. The 95% confidence intervals (CIs) of the AUCs are shown in parentheses. The *blue line* indicates ROC based on TMA lyases (CutC/D, YeaW/X), with its corresponding ACVD probability shown in Supplementary Data 8 and Supplementary Table 3. The best cutoff points were marked on the ROCs

levels, was biosynthesis of phosphatidylethanolamine (PE) (Supplementary Fig. 2c), a membrane lipid that may promote coagulation[33].

**Influence of drugs on the gut microbiota**. Studies comparing individuals with T2D treated or not with metformin identified a higher level of *E. coli* and a lower level of *Intestinibacter* in the metformin-treated individuals, while confirming reduced levels of

butyrate-producing bacteria such as *Roseburia* spp., *F. prausnitzii*, and unnamed *Clostridiales* in the untreated individuals compared to healthy controls[5, 6, 34, 35]. Moreover, cross-sectional as well as intervention studies have demonstrated a significant impact on the gut microbiota by proton-pump inhibitors[36–38]. Here, we evaluated potential complications from drug use in this ACVD cohort.

PERMANOVA[39] identified a significant influence on the abundances of gut microbial genes by the anticoagulant fondaparinux, the T2D drug acarbose, the beta adrenergic receptor antagonist metoprolol, and to a lesser extent atorvastatin (known as Lipitor) (these four drugs were the only drugs with *P* < 0.1, but all had *P* > 0.01, Supplementary Table 2). For all drugs, random-forest classifiers reached a higher AUC (and a larger Youden's index) for distinguishing between ACVD patients with no drug treatment and healthy controls, than between ACVD with and without drugs (Fig. 5). With the exception of metoprolol, the AUC and Youden's index were also higher between ACVD patients treated with the drug and healthy controls than between ACVD with and without drug treatment (Fig. 5). When two drugs were analyzed together instead of single drug analysis, random-forest classifiers again reached a higher AUC for distinguishing between ACVD patients with no drug treatment and healthy controls, than between ACVD with and without the drugs (Supplementary Data 4). For combinations such as either acarbose or atorvastatin, the AUC was lower between ACVD patients treated with the drugs and healthy controls than between ACVD patients without the drugs and healthy controls (Supplementary Data 4), i.e., the medication weakened the disease signal, meaning an even more significant difference would be expected if the cohort was free of medication. Thus, these results suggest that ACVD status, and not current drug use, is the major distinguishing feature in this cohort. Still, drug treatment may to varying extent affect the composition of the gut microbiota and thus constitute a confounding factor. The possible effects of different drug use on the gut microbiome composition remain to be explored in intervention trials in larger cohorts.

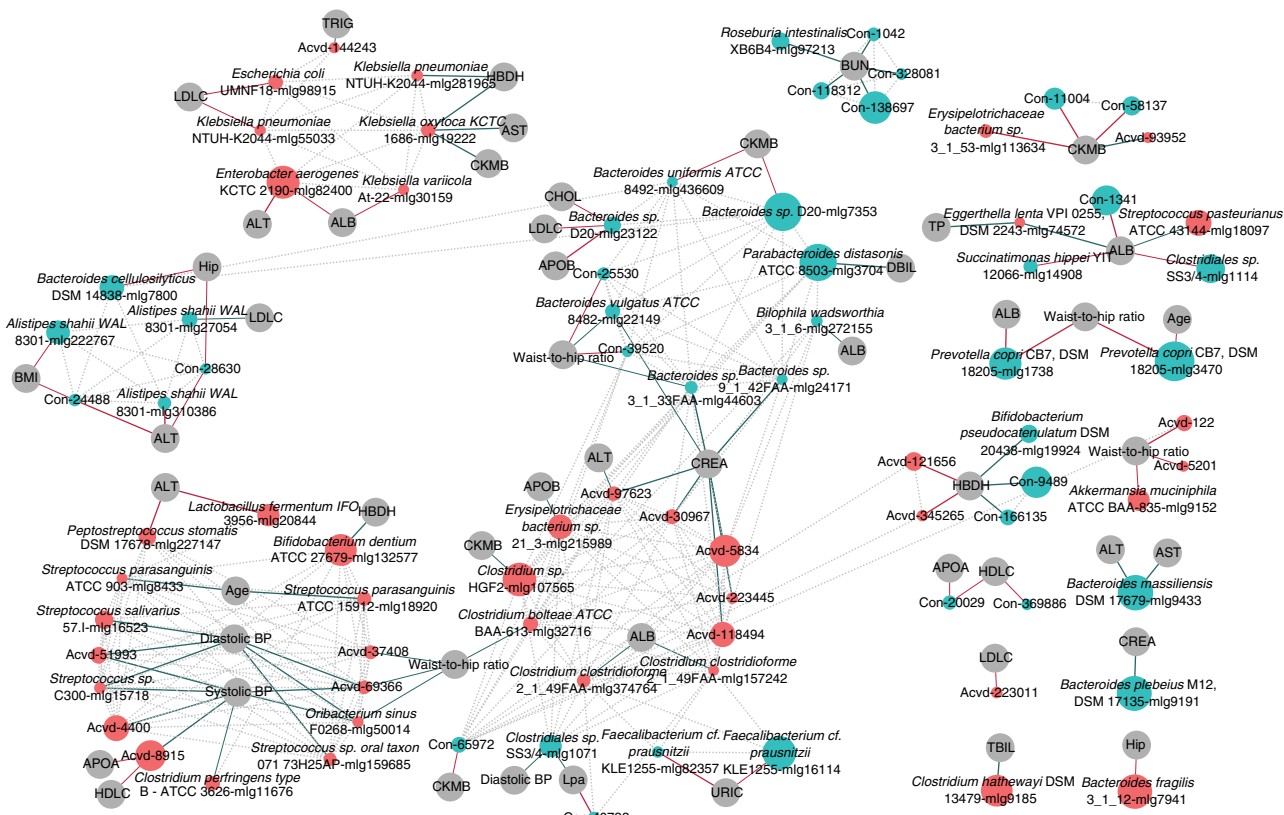

**Fig. 4** Associations between ACVD-enriched or depleted MLGs and clinical indices. Differentially enriched MLGs (*q*-value <0.05, FDR-controlled Wilcoxon rank-sum test, Fig. 2, Supplementary Data 3) were analyzed for associations with clinical indices (Supplementary Data 1). ACVD-enriched MLGs are represented by *red circles*, and control-enriched MLGs are represented by *cyan circles*. The size of each circle indicates the number of genes in an MLG (100–3723, detailed in Supplementary Data 3). Associations were defined as permutational Wilcoxon rank-sum test *P*-value for Spearman correlation <0.05, |Spearman's ccl ≥ 0.2 and selected by an MLG-based RFCV model for the clinical index. The *thicker lines* had a stronger association of |Spearman's ccl > 0.25. *Green lines* indicate positive association, and *red lines*—negative associations. *Dotted grey lines* indicate correlations between MLGs in the control samples, as shown in Fig. 2. Clinical indices: age, ALB (albumin), ALT, APOA, APOB, AST, BMI (body mass index), BUN (blood urea nitrogen), CHOL (cholesterol), CKMB, CREA (creatinine), DBIL (direct bilirubin), diastolic BP (diastolic blood pressure), HBDH, HDLC (high-density lipoprotein cholesterol), hip (hip circumference), LDLC (LDL cholesterol), Lpa (lysophosphatidic acid), systolic BP (systolic blood pressure), TBIL, TP (total protein), TRIG (triglyceride), URIC (uric acid), and waist-to-hip ratio

Furthermore, we investigated potential influence of drug use on specific members of the gut microbiome using MaAsLin. The relative abundances of three MLGs positively correlated with metoprolol use and two MLGs positively correlated with atorvastatin, but none of the MLGs fulfilled the criteria for species annotation (Supplementary Fig. 3, Supplementary Data 3). The drug use therefore did not perceivably complicate our elucidation of bacterial species or strains associated with ACVD. The positive associations with control-enriched MLGs might actually represent previously overlooked mechanisms of action of metoprolol and atorvastatin (Supplementary Fig. 3).

**Functional alterations in the ACVD gut microbiome.** To increase the insight into functional changes within the ACVD gut microbiome, we determined to what extent different KEGG pathways and modules were enriched in the gut microbiota of patients compared to controls (Figs. 6 and 7, Supplementary Fig. 4a and b, Supplementary Data 5–7). The samples from ACVD patients displayed higher potential for transport of simple sugars (phosphotransferase systems (PTS)) and amino acids, whereas the potential for biosynthesis of most vitamins was lower (Figs. 6 and 7, Supplementary Data 6 and 7). Folate is known to play a role in cardiovascular disease due to its function in homocysteine metabolism[40], and we observed a reduced potential

for the synthesis of tetrahydrofolate and altered potential for homocysteine metabolism in the gut microbiome of ACVD patients compared with controls (Fig. 6, Supplementary Fig. 4a and b). The ACVD microbiome moreover exhibited reduced potential for metabolizing glycans including glycosaminoglycans (Fig. 7). Consistent with the enrichment of *Enterobacteriaceae* in ACVD (Figs. 1 and 2), the module comprising genes required for the synthesis of the O-antigen of lipopolysaccharides (LPS) was enriched in ACVD samples, whereas the lipid A synthesis module was relatively depleted, most likely due to a lower level of the otherwise abundant Gram-negative genus *Bacteroides* (Figs. 1 and 6). The latter represents non-inflammatory penta-acylated lipid A-producing species[41]. According to the virulence factor database (VFDB)[27], the ACVD samples were also significantly enriched in virulence factors compared to the control samples (Supplementary Fig. 6c). The potential for the metabolism of glycerolipids and degradation of fatty acids was elevated in ACVD (Fig. 7), whereas the potential for synthesis of the anti-inflammatory short-chain fatty acid (SCFA) butyrate was lower (Fig. 8a). Similarly, one module involved in propionate synthesis was less abundant in ACVD patients compared with controls. No significant changes were observed for pathways involved in the synthesis of acetate (Fig. 8a, Supplementary Data 5–7). Gut microbial enzymes involved in formation of TMA, the precursor for the proatherogenic metabolite TMAO[11–14, 42], were enriched

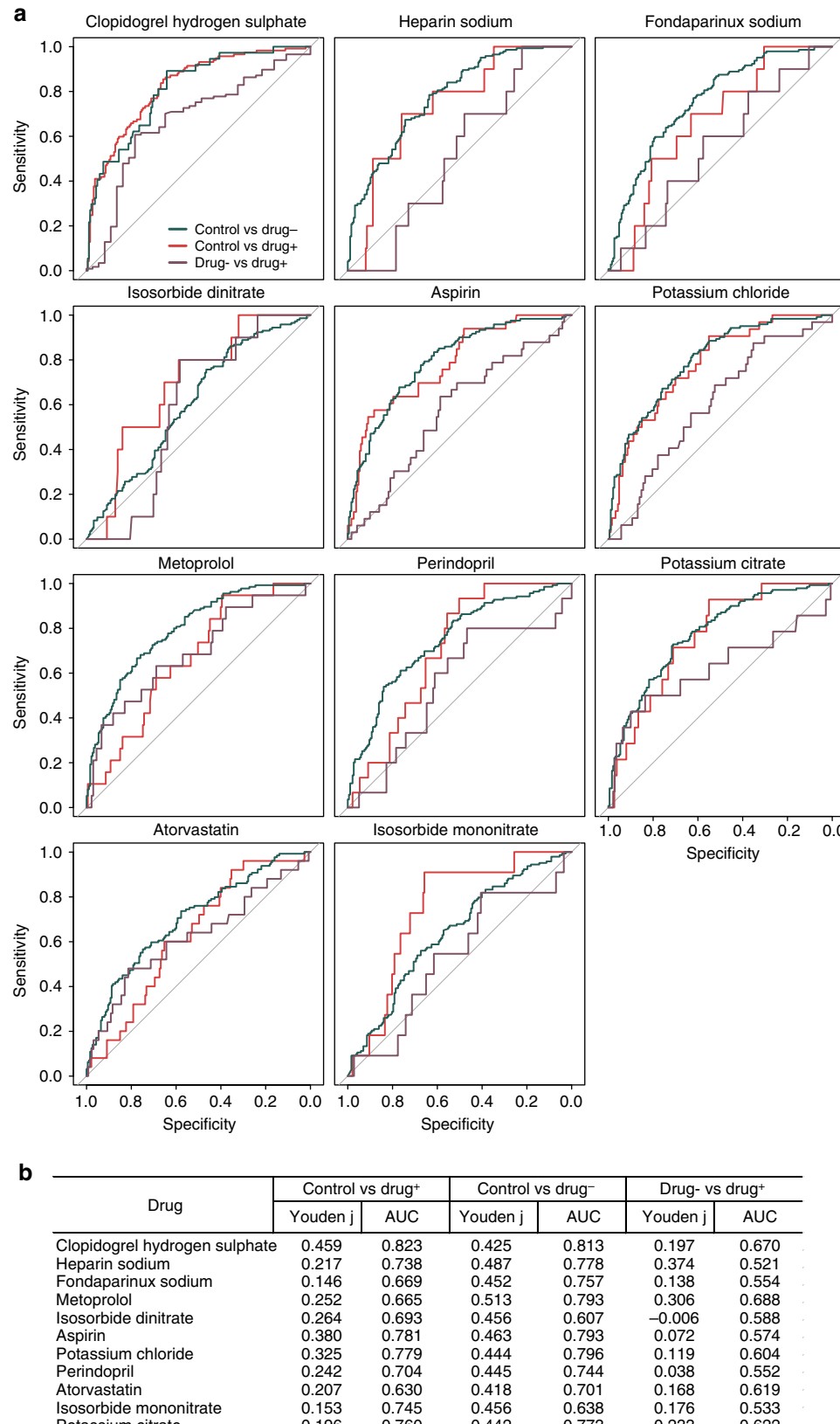

**Fig. 5** Performance of random forest classifiers for ACVD samples stratified by medication. **a** ROC for two-way classification models of ACVD patients with and without the drug and healthy controls. Fivefold cross-validation repeated five times. Numbers of individuals treated with each drug are given in Supplementary Data 2. **b** AUC and Youden's index for the ROC plots in **a**

| Drug | Control vs drug+ | | Control vs drug− | | Drug- vs drug+ | |
|---|---|---|---|---|---|---|
| | Youden j | AUC | Youden j | AUC | Youden j | AUC |
| Clopidogrel hydrogen sulphate | 0.459 | 0.823 | 0.425 | 0.813 | 0.197 | 0.670 |
| Heparin sodium | 0.217 | 0.738 | 0.487 | 0.778 | 0.374 | 0.521 |
| Fondaparinux sodium | 0.146 | 0.669 | 0.452 | 0.757 | 0.138 | 0.554 |
| Metoprolol | 0.252 | 0.665 | 0.513 | 0.793 | 0.306 | 0.688 |
| Isosorbide dinitrate | 0.264 | 0.693 | 0.456 | 0.607 | −0.006 | 0.588 |
| Aspirin | 0.380 | 0.781 | 0.463 | 0.793 | 0.072 | 0.574 |
| Potassium chloride | 0.325 | 0.779 | 0.444 | 0.796 | 0.119 | 0.604 |
| Perindopril | 0.242 | 0.704 | 0.445 | 0.744 | 0.038 | 0.552 |
| Atorvastatin | 0.207 | 0.630 | 0.418 | 0.701 | 0.168 | 0.619 |
| Isosorbide mononitrate | 0.153 | 0.745 | 0.456 | 0.638 | 0.176 | 0.533 |
| Potassium citrate | 0.196 | 0.760 | 0.442 | 0.773 | 0.233 | 0.622 |

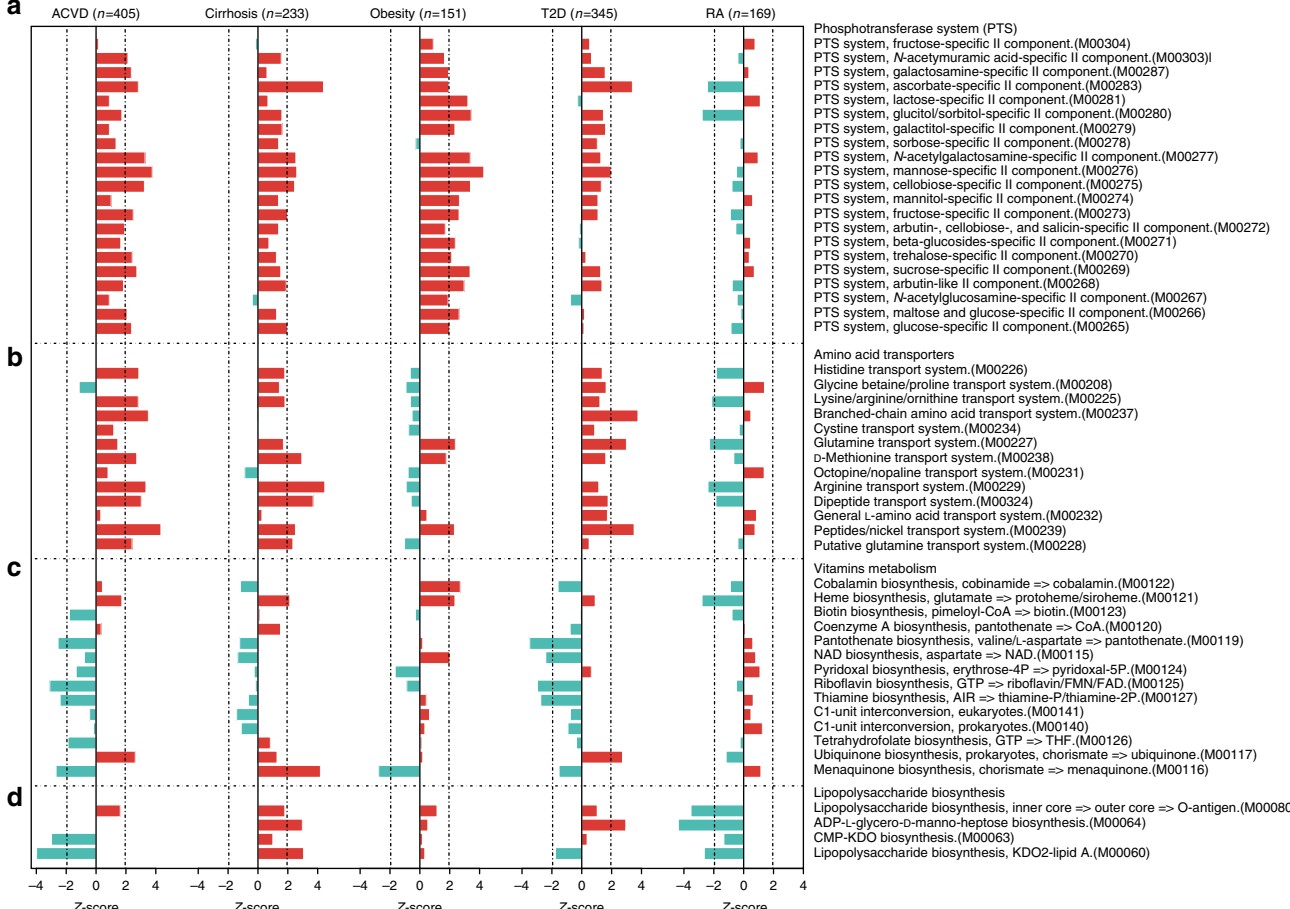

**Fig. 6** Alterations in gut microbial functional modules in ACVD and other diseases. **a** PTS transport systems. **b** Amino acid transporters. **c** Vitamin metabolism. **d** LPS biosynthesis. *Red*, case-enriched; *cyan*, control-enriched, compared within each disease cohort (ACVD, total $n = 405$ including 218 cases; cirrhosis, total $n = 231$ including 120 cases; obesity, total $n = 151$ including 72 cases; T2D, total $n = 345$ including 171 cases; RA, total $n = 169$ including 95 cases). *Dashed lines* indicate a reporter score of 1.96, corresponding to 95% confidence in a normal distribution

in the ACVD samples compared to healthy controls, especially for YeaW/X (Fig. 8b). The gut microbiome in individuals with ACVD might also produce more formate, judging from the enrichment of pyruvate-formate lyase (EC2.3.1.54, K00656) in the ACVD samples compared to controls (Supplementary Data 5). Pyruvate-formate lyase is a key enzyme for formate biosynthesis[43]. Formate has been implicated in hypertension[44, 45], and might contribute to other ACVD-related functions including methanogenesis and one-carbon metabolism.

With this large cohort of ACVD, we next explored common changes in the gut microbiome of other linked diseases; liver cirrhosis, the other CMDs, obesity, and T2D, as well as in the non-linked autoimmune disorder RA. Published metagenomic shotgun-sequencing data for liver cirrhosis[46] (120 cases and 111 controls), obesity[47] (72 cases and 79 controls), T2D[4] (171 cases and 174 controls), and RA[25] (95 cases and 74 controls) were mapped to the same reference gene catalog and analyzed for functional potentials using the same pipeline. Due to substantial differences in the gut microbiome among different populations[5, 18, 48], only data from Chinese cohorts were used, although high-quality data on T2D and obesity cohorts also exists from European countries[3]. Compared with controls, differences in functional potential within the gut microbiome of the two other CMDs T2D and obesity were similar to those observed in ACVD patients. By contrast, in patients with RA, we did not observe the striking enrichment of many modules of the PTS, instead only a few modules were enriched in controls compared

to cases. Further, in RA patients modules involved in glutamine and arginine transport and modules involved in LPS biosynthesis were enriched in controls compared to cases, in contrast, modules for host glycan degradation were enriched in RA patients (Figs. 6 and 7, Supplementary Data 6). These results point to an intricate balance between the gut microbiome in metabolic and autoimmune diseases.

Compositionally, we also noted common features in the three CMDs and cirrhosis. The ACVD-based classifier (Fig. 3) contained features such as *Streptococcus* spp. that were also abundant in the cirrhosis patients, however, a number of *Eggerthella* MLGs were not enriched in the individuals with cirrhosis (Supplementary Fig. 5). Moreover, the majority of *E. coli* MLGs enriched in individuals with ACVD did not show significant differences between cases and controls within the other diseases (Supplementary Fig. 5), indicating that both common and unique features of the gut microbiome could be identified in ACVD.

In addition to bacteria and archaea, a number of bacteriophages were more abundant or less abundant in the gut microbiome of individuals with ACVD compared to healthy controls (Supplementary Fig. 6), and several of these were also highly enriched in cirrhosis patients. In contrast, none of these bacteriophages displayed a significant difference in abundance in the RA cohort (Supplementary Fig. 6). The known hosts for the ACVD-enriched phages mostly included bacteria from the family *Enterobacteriaceae* or the genus *Streptococcus* (Supplementary Figs. 6 and 7). Judging from the positive correlations between the

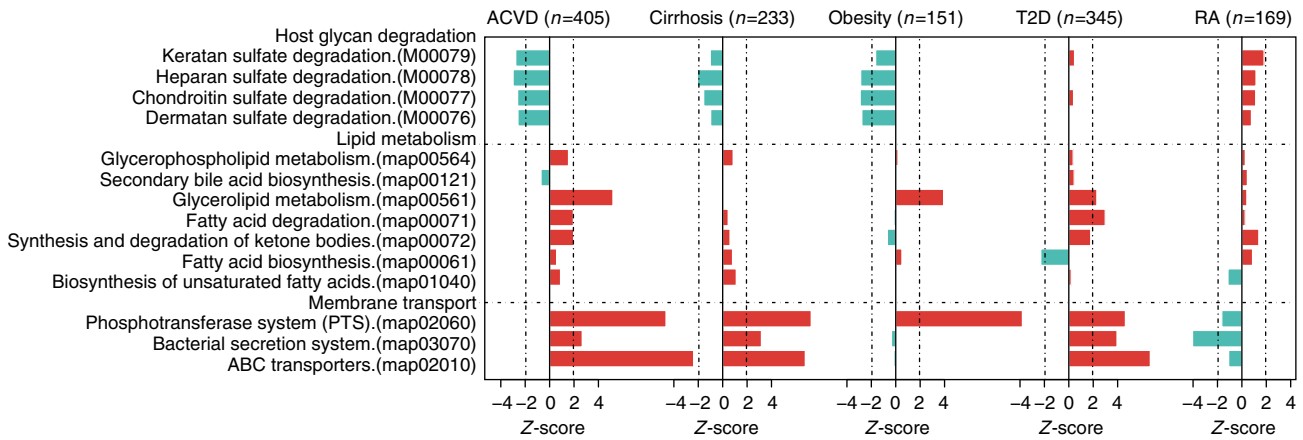

**Fig. 7** Differential enrichment of membrane transport pathways, lipid metabolism pathways, and modules for host glycan degradation. X-axis represents reporter score. *Red*, case-enriched; *cyan*, control-enriched, compared within each disease cohort (ACVD, total $n = 405$ including 218 cases; Cirrhosis, total $n = 231$ including 120 cases; Obesity, total $n = 151$ including 72 cases; T2D, total $n = 345$ including 171 cases; RA, total $n = 169$ including 95 cases). *Dashed lines* indicate a reporter score of 1.96, corresponding to 95% confidence in a normal distribution

phages and their host bacteria (Supplementary Fig. 8), we expect most of them to be integrated rather than present as free viral genomes, while formal detection of any lytic populations would require isolation of phage particles.

**Microbiome-based identification of CMDs and cirrhosis.** Despite the generally high resemblance of the gut microbiome functionality within the three CMDs as well as liver cirrhosis, we wondered if it might be possible to identify MLGs and functionalities that could distinguish these patient groups. Among the top 50 determining species, the ACVD-enriched MLGs such as *S. anginosis*, *S. vestibularis*, and *L. salivarius* were identified to be most important for the model, while ACVD-depleted MLGs such as *F. cf. prausnitzii*, *Bacteroides cellulosilyticus*, *R.intestinalis*, and MLGs not significantly different in the ACVD cohort were also selected as important features (Supplementary Fig. 9a). A model could also be constructed based on KEGG Orthology (KO) modules using cross-validated group LASSO (Supplementary Fig. 9b). The top-ranked module for the CMDs and liver cirrhosis was the ACVD-microbiome depleted DegS-DegU two component system, which can trigger biofilm formation in response to impeded flagellar rotation[49]. The ACVD-microbiome depleted taurine transport system was also important for this classifier (Supplementary Figs. 2c and 9b, Supplementary Data 6), consistent with the importance of taurine for cardiometabolic health.

Besides the converging features (Supplementary Fig. 9a and b), it was also possible to classify an individual into one of the CMDs or cirrhosis based on distinguishing features of the gut microbiome. *Veillonella* spp. were most important for identifying individuals with liver cirrhosis, *B. vulgatus* for T2D, and *Dorea longicatena* for obesity (Supplementary Fig. 9c). The two unclassified species (Acvd-4400 and Acvd-69366) separated ACVD from the remaining CMDs. This is to our knowledge the first multi-disease classifiers using the gut microbiome to identify unique features for each disease.

## Discussion
In the present study, we exploited metagenomic shotgun-sequencing data from a large cohort of individuals with ACVD enabling a comprehensive comparison across the linked CMDs, obesity, and T2D, as well as to liver cirrhosis, and the non-linked autoimmune disease RA. All 1250 samples were analyzed in the same manner, eliminating technical biases in MWAS between studies[3]. This is also the first time that such a cross-disease cohort

was profiled according to a comprehensive high-quality reference gene catalog that includes both cultivated and uncultivated microbes[18], and further explored for inference of functionality.

We observed that the gut microbiome collectively is less fermentative and more inflammatory in patients with CMDs and liver cirrhosis, in contrast to that of patients with the autoimmune disease RA, suggesting that the restoration of a few members of the healthy gut microbiome might alleviate or at least reduce the risk for multiple CMDs and possibly liver cirrhosis. Alternatively, the altered microbiome of CMD patients might be a read-out of the systemic inflammatory component of CMDs and cirrhosis, hence illustrating an ongoing interaction between the gut microbiome composition and the inflammatory status of these diseases that is detectable as a CMD/cirrhosis-specific gut microbiome signature. If so, then the restoration toward a healthy microbiome might involve concomitant anti-inflammatory treatments. Mechanistically, a better understanding of the ecology of the gut microbiota would also be critical for understanding its role in the ever-increasing list of microbiota-associated diseases. Although the gut microbiome of liver cirrhosis, colorectal cancer, RA, and ACVD patients have all showed an increase in the abundance of oral bacteria[3], the set of oral bacterial species were, however, observed to differ, and only RA and ACVD have been epidemiologically associated with periodontitis.

Besides the common denominators identified to characterize the gut microbiome in patients with CMDs and cirrhosis, the gut microbiome of each disease also exhibited unique features in functional capacity, and in species and gene compositions. Although several MLGs were annotated to known species, they varied to different degrees from sequenced reference genomes. Isolation of the disease-relevant strains, and/or engineering of reference strains to contain disease-relevant genes, would be important first steps before taking the bacteria or communities to test in animal models. For ACVD, in particular, the myriad of LPS structures, TMA-lyase activities, folate-utilizing enzymes, and specific proportions of each SCFA are all among the important functions to tease out both in vitro and in vivo. Our metagenomic shotgun-sequenced cohorts have provided information on the species and genomic functionality of these to further the understanding of cause and consequence in CMDs and liver cirrhosis.

## Methods
**Cohorts and clinical biomarkers for ACVD.** Samples from 405 Chinese subjects[50], including 218 individuals with ACVD and 187 control subjects, were

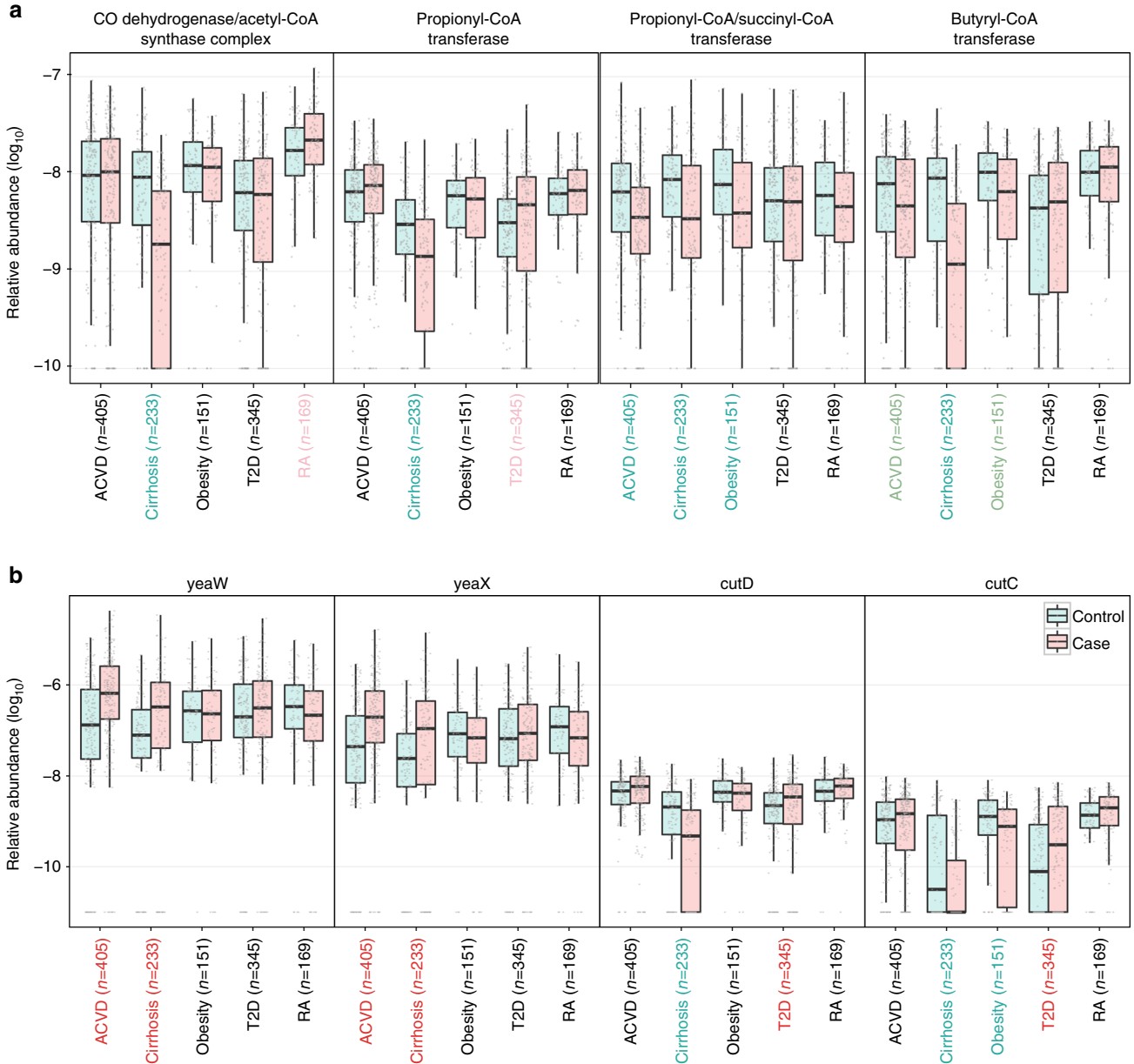

**Fig. 8** Abundance differences in specific gut microbial KOs in ACVD and other diseases. **a** Enzymes for production of SCFAs. **b** Enzymes for TMA production. CutD is an activator for CutC. The cohort names were colored according to the direction of enrichment, i.e. *green* and *red* for control- and case-enriched, respectively within each disease cohort (ACVD, total $n = 405$ including 218 cases; Cirrhosis, total $n = 231$ including 120 cases; Obesity, total $n = 151$ including 72 cases; T2D, total $n = 345$ including 171 cases; RA, total $n = 169$ including 95 cases). Wilcoxon rank-sum test, *P*-value < 0.01; *light green* and *light red*, Wilcoxon rank-sum test $0.01 \leq P$-value < 0.05; black, *P*-value $\geq 0.05$. *Boxes* represent the median and interquartile ranges (IQRs) between the first and third quartiles; *whiskers* represent the lowest or highest values within 1.5 times IQR from the first or third quartiles. *Circles* represent all data points

collected at the Medical Research Center of Guangdong General Hospital. Individuals with ACVD showed clinical presentations of stable angina, unstable angina, or acute myocardial infarction (AMI) (Supplementary Data 1). ACVD diagnosis was confirmed by coronary angiography, and individuals that had ≥50% stenosis in single or multiple vessels were included. All patients were ethnic Han Chinese with no known consanguinity, aged 40–80 years old. The exclusion criteria included ongoing infectious diseases, cancer, renal, or hepatic failure, peripheral neuropathy, stroke, as well as use of antibiotics within 1 month of sample collection. All the healthy control individuals enrolled were free of clinically evident ACVD symptoms at the time of the medical examination. Demographic data and cardiovascular risk factors were collected by a questionnaire. Individuals with peripheral artery disease, known coronary artery disease or myocardial infarction, cardiomyopathy, renal failure, peripheral neuropathy, systemic disease, and stroke were excluded.

Fresh feces, early morning urine, and blood samples of each subject were collected the first morning after admission to the hospital. All collected samples

were frozen on dry ice within 30 min, and stored in −80 °C freezers before further analysis.

Blood samples for clinical chemistry analyses were taken after an overnight fast for at least 10 h. Fasting or 2 h glucose, serum alanine aminotransferase (ALT), aspartate aminotransferase (AST), alkaline phosphatase (ALP) and γ-glutamyl transpeptidase (GGT), TBIL, creatinine, uric acid, lipid profile, including triglycerides (TG), TC, high-density lipoprotein cholesterol, and LDL cholesterol were measured using an autoanalyzer (Beckman Coulter AU5800). HBA1C was measured by high-pressure liquid chromatography.

The study was approved by the Medical Ethical Review Committee of the Guangdong General Hospital and the Institutional Review Board at BGI-Shenzhen. Informed consent was obtained from all participants.

**DNA extraction from fecal samples**. Fecal samples were thawed on ice and DNA extraction was performed using the Qiagen QIAamp DNA Stool Mini Kit (Qiagen)

according to manufacturer's instructions. Extracts were treated with DNase-free RNase to eliminate RNA contamination. DNA quantity was determined using NanoDrop spectrophotometer, Qubit Fluorometer (with the Quant-iTTMdsDNA BR Assay Kit), and gel electrophoresis.

**DNA library construction and sequencing of fecal samples.** DNA library construction was performed following the manufacturer's instruction (Illumina). We used the same workflow as described previously[4] to perform cluster generation, template hybridization, isothermal amplification, linearization, blocking and denaturation, and hybridization of the sequencing primers. We constructed one paired-end (PE) library with insert size of 350 bp for each sample, followed by a high-throughput sequencing with PE reads of length 2 × 100 bp. High-quality reads were obtained by filtering low-quality reads with ambiguous "N" bases, adapter contamination, and human DNA contamination from the Illumina raw reads, and by trimming low-quality terminal bases of reads simultaneously[4]. Over 90% of the reads remained after the filtering for low quality and then human reads (mapped to the hg19 reference) (55.2 million reads per sample, Supplementary Data 2).

**PCA and dbRDA.** Principle component analysis (PCA) was performed on the genus level as previously described[4]. dbRDA[51] was performed using Bray–Curtis distance also on the genus level. Similar to ref. 6, dbRDA was used as a supervised complement to PCA, to better present the difference between ACVD and control samples. It is part of the vegan package of R 3.3.0.

**KEGG analysis.** Differentially enriched KO pathways or modules were identified according to their reporter score from the Z-scores of individual KOs (KEGG database release 59.0, genes from animals and plants removed)[52–54]. One-tail Wilcoxon rank-sum test was performed on all the KOs that occurred in more than five samples and adjusted for multiple testing using the Benjamin–Hochberg procedure[55]. The Z-score for each KO could then be calculated. Absolute value of reporter score = 1.6 or higher (95% confidence on either tail, according to normal distribution) could be used as a detection threshold for significantly differentiating pathways.

Sequences of SCFA-producing enzymes were retrieved as previously described[56]. Genes in the reference gut microbiome gene catalog[18] were identified as these enzymes (best match according to BlastP, identity >35%, score >60, $E < 1e{-}3$), and their relative abundances could then be determined accordingly.

The TMA-lyases were analyzed in the same manner. We retrieved the sequences of the enzymes from Uniprot (X8HTY7, X6QEX0, A0A0E2Q854, W3YJY9, U2TK70, A0A0M3KL45, A0A0M3KL44, A0A0M4MQL2, and A0A0M4N7P9 for the choline TMA-lyase CutC[57]; D0C9N6 and D0C9N8 for the carnitine TMA-lyase CntA/B[58] and from NCBI (219868924, 78220727 and 219868925, 342906634 for CutC/D; 169889293 and 169889294 for the promiscuous carnitine TMA-lyase YeaW/X (whose in vitro substrates included γ-butyrobetaine, ʟ-carnitine, choline, and betaine))[59]. CntA/B was not found in the gut microbiome, consistent with the original report on its presence in Acinetobacter[58].

**Virulence factors.** Virulence factors were analyzed according to VFDB[27] (2585 proteins as of 16 August 2016). Genes in the reference gut microbiome gene catalog[18] were identified as these virulence factors (best match according to BlastP, identity >35%, score >60), and their relative abundances could then be determined accordingly.

**Metagenome-wide association study.** MWAS was performed as previously described[3] on the ACVD cohort. The clean reads (after removing low quality reads and host reads) were mapped to the 9 879 896 genes in the reference catalog[18]. Genes detected in <10 samples were removed, resulting in a set of 3 694 132 genes and gene abundance profiles. These genes were then clustered into MLGs according to their abundance variation across all samples[4]. The relative abundance of each MLG in each sample was summed from the relative abundance of their constituent genes, as previously described[4]. The 2982 MLG that contained over 100 genes were tested for enrichment or depletion in individuals with ACVD (218 ACVD cases and 187 controls) according to the non-parametric method, Wilcoxon rank-sum test, and controlled for multiple testing (q-value <0.05)[55, 60]. In addition, we analyzed the associations of disease status with genes, MLGs, and functions.

MLGs were further clustered according to Spearman's correlation between their abundances in all samples regardless of case–control status, and the co-occurrence network was visualized by Cytoscape 3.4.0.

Taxonomic assignment of the MLGs was performed according to the taxonomy of their constituent genes, which were aligned to the NCBI reference genomes (>50% of the genes in one MLG) as previously described[4]. For each MLG with a tentative species annotation, the percentage of genes covered and the average identity were shown for the top three reference genomes (Supplementary Data 3).

**Quantification of bacteriophages.** Genes corresponding to bacteriophages in the reference gut microbiome gene catalog[18] were identified through alignment to NCBI (5781 viral genomes as of 8 August 2016; best match according to BlastN,

identity >65%, score >60). A total of 1095 viruses were identified, and their relative abundances were summed from the relative abundances of their genes.

**Gene content within species.** Species annotation of the ACVD-enriched or -depleted MLGs from the MWAS analysis (q-value <0.05) were used to retrieve reference genome sequences of the bacteria. PanPhlAn[26] was then used to build a unique gene set for each species, map the sequencing reads against the gene set, and determine the presence or absence of each bacterial species in each sample. For each species, genes that differed in occurrence between ACVD cases and controls were identified by Fisher test, q < 0.05. These genes were then functionally annotated according to the KEGG database[31] (best match according to BlastP, identity >35%, score >60), as well as the VFDB database[27] for virulence factors, UniProt and NCBI for TMA-lyases and bacteriophages (5781 viral genomes, downloaded on 8 August 2016) as was detailed above.

**Feature selection using RFCV or group LASSO.** Fivefold cross-validation was performed on a random forest model (R 3.3.0, randomForest 4.6-12 package) using the MLG abundance profile of the samples. The cross-validation error curves (average of five test sets each) from five trials of the fivefold cross-validation were averaged, and the minimum error in the averaged curve plus the standard deviation at that point was used as the cutoff[25, 53]. All sets (≤25) of MLG markers with an error less than the cutoff were listed, and the set with the smallest number of MLGs was chosen as the optimal set. The probability of ACVD was calculated using this set of MLGs and an ROC was drawn (R 3.3.0, pROC package). The ROC and ACVD probability based on TMA-lyases were plotted using the same R version and package (R 3.3.0, pROC package).

CMD/cirrhosis-specific MLGs were also selected by a random forest model, with downsampling to account for the unequal sample size across diseases.

To select KO modules characteristic of ACVD or the other diseases, group LASSO[32] was used (R 3.3.0, SGL package). Specifically, logistic regression with 10-times bootstrapping was used for identifying disease samples from controls; linear regression was used for the clinical indices, all based on the relative abundances of KO modules in the samples.

**Associations between the microbiome and clinical indices.** For ACVD-enriched or -depleted MLGs (Q < 0.05), we investigated their correlation with clinical indices. First, a many (MLGs) to one (one index) fivefold cross-validation random forest selection (RFCV) was done to select a few MLGs for each index. We further selected the correlations with Spearman's permutational P-value <0.05 and Spearman's |cc| ≥ 0.2.

**Assessing potential influences of medication.** Potential influences of medication on the ACVD-associated gut microbiome were analyzed by three complementing methods.

Cross-validated random forest (RFCV) models were used in the same manner as in the construction of the ACVD classifier.

PERMANOVA[39] was performed on the MLG abundance profile of all samples to assess impact from each of the factors listed (Supplementary Table 2). We used Jenson–Shanon divergence and 999 permutations in R (3.0.2, vegan package)[61].

Multivariate association with linear models (MaAsLin), a linear modeling system adapted for microbial community data (http://huttenhower.sph.harvard.edu/maaslin)[62] was applied to the data according to the authors' instructions. Default parameters were used, except that MLGs of low abundance were not considered (relative abundance <1e−7, Supplementary Data 3).

**Data availability.** The metagenomic shotgun-sequencing data for all samples have been deposited in the European Bioinformatics Institute (EBI) database under the accession code ERP023788. The EBI IDs for each sample are shown in Supplementary Data 2. The authors declare that all other data supporting the findings of the study are available within the paper and its Supplementary Information files, or from the corresponding authors upon request.

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

## Acknowledgements

This research was supported by the Natural Science Foundation of China (no. 81373486 and 81673514), the National Key Research and Development Program of China (No. SQ2017YFSF090209), the Science and Technology Development Projects of Guangdong Province, China (no. 2016B090918114), the Shenzhen Municipal Government of China (JSGG20160229172752028 and JCYJ20160229172757249), and the Fund for Science and Technology Development (FDCT) from Macao (grant 077/2014/A2). We gratefully acknowledge colleagues at BGI-Shenzhen for DNA extraction, library construction, sequencing, and discussions.

## Author contributions

S.-L.Z., R.L., J.-Y.C., Q.-S.G., Z.-W.Z., and K.H. performed the clinical diagnosis for ACVD, collected samples, and provided metadata. Z.L. coordinated the sample processing at BGI. Z.J., H.X., S.L., S. Liang., H.Z., Z.L., Y.G., D.Z., Z.S., Z.F., J.L., L.X., J.L., X.L., F.L., H.R., Y.H., Y.P., G.L., B.W., and B.D. performed the bioinformatic analyses, and prepared figures and tables. H.Y., Jian W., Jun W., X.X., X.L., and Y.H. provided resources and commented on the results. X.Z. and G.N. helped in providing the RA and obesity data, respectively, for comparison with ACVD. Q.F. and H.Z. wrote early drafts, and H.J., Z.J., L.M., S.B., and K.K. wrote the final version and led significant reorganization of the study. All authors contributed to the revision of the manuscript.

## Additional information

**Competing interests:** The authors declare no competing financial interests.

