## [Peer Review File · Nature Communications]

Editorial Note: this manuscript has been previously reviewed at another journal that is not operating a transparent peer review scheme. This document only contains reviewer comments and rebuttal letters for versions considered at Nature Communications. Mentions of the other journal have been redacted.

Reviewers' comments:

Reviewer #2 (Remarks to the Author):

The authors have made an earnest effort to address the criticisms of all the reviewers and while there are still significant weaknesses, including low power and lack of causal inference, the paper also has important strengths:

1. This is the first metagenome wide association study (MWAS) for ACVD and can be used as a reference for future investigations on the role of microbiome composition in ACVD.
2. The number of individuals in the study is similar or greater than that of previously published MWAS in human disease cohorts.
3. Abundance of specific bacterial species and alterations in functional potential of the metagenome were correlated with ACVD.
4. Authors tested for the potential interaction of drugs on gut microbiome composition.
5. The authors identified novel commonalities and differences between the ACVD metagenome and metagenomes of other diseases.

There are some details missing regarding sequencing depth and coverage. In the methods, the authors write that each sample was sequenced with "around 30 million paired-end reads." The mean number of reads per sample and range should be disclosed in the results section. The sequencing coverage across the metagenome is not discussed, but could have important implications on the validity of results.

Reviewer #3 (Remarks to the Author):

The authors have attempted to address the numerous issues by removing data (metabolomics) and expanding sample numbers, or exploring microbiota taxa associations with alternative cardiometabolic conditions and trying to demonstrate unique associations between those identified in their expanded cohort for ACVD, vs alternative comorbidities.

While it is appreciated that the authors have attempted to address concerns, they did so by providing virtually identical sorts of associative data, and fail to address the fundamental challenges and limitations of a purely associative exercise, which this study still remains.

The data remain associative and likely much will not replicate with an independent prospective cohort. As was recently shown for metagenomics analyses revealing associations with T2DM, associations previously reported as being signatures for T2DM that were even "validated" by others, ended up being associated with medication use (Metformin).

Thus, even if it associations reported in metagenomics data were to replicate, with the analyses performed, there are too many comorbidities and medications that are not adequately adjusted for. The "controls" are essentially healthy. - not propensity match (age, sex, comorbidities matched) subjects who are followed prospectively to show they do vs do not experience development of ACVD or adverse events.

While 5-fold folding sorts of analyses help to try and identify robust and reproducible associations of clusters of phenotypes, the approaches taken still have inherent limitations including:

- 1) being fundamentally associative only in nature, and failing to discriminate with other comorbidities.
- 2) Lacking in use of independent non-overlapping validation prospective cohort
- 3) Lacking in ability to identify microbiota, pathways or metabolites that are mechanistically or causally associated with the disease process.
- 4) Moreover, the removal of the metabolomics data deteriorates the validity of the inferred Kegg pathway "functional" analyses. As the authors acknowledge, these are "hypothesis generating" exercises and lack validation. At least performance of validated stable isotope dilution LC/MS/MS analyses for candidate significant metabolites would have allowed for independent validation of the inferred functional association. So the removal of the data, rather than performance of the experimental studies and analyses, further weakened the manuscript, not improved it.
- 5) Kegg pathway analyses to infer function remains inference. Assigned enzyme activities are based solely on prediction based on sequence homology. Until actual biochemical data are experimentally tested for and verified, it is hypothesis only.

Reviewer #4 (Remarks to the Author):

This study describes a metagenomic investigation of the microbiome associated with atherosclerotic cardiovascular disease (ACVD) based on 408 fecal samples collected from Chinese individuals and metagenomically sequenced exclusively for this study. The dataset is then integrated with 845 samples from previously published studies. The sample size is in line with other metagenomic studies on the association between the microbiome and other diseases. Results showed the depletion of Bacteroides and Prevotella in ACVD patients, and enrichment of Escherichia and Streptococcus. Also Enterobacter aerogenes and Klebsiella pneumoniae are enriched in disease and these bacteria possess genes for TMA lysase whose activity is linked to atherosclerosis. Random forest classifier identified bacterial species that can be treated as features to discriminate ACVD samples from controls subjects. From a functional potential point of view, the ACVD microbiome share some similarity to the obesity and T2D one with an increase of PTS pathways and a decrease in the vitamin biosynthesis pathways (THF).

Overall, I think this is an interesting study which provides relevant results and data. However, I have two major issues to point out. The first is the potential problem of having treatment as confounder for disease. The second is the overwhelming amount and types of analyses that have been performed which are not always focused toward the main message.

1. Drugs as confounding factor. The authors try to address this potential problem in their revision. They show in the section "Influence of drugs on the gut microbiota" that drugs do have an influence on the gut microbiome but they somehow conclude that this effect is smaller than that of the disease. I can partially agree with their conclusion when considering each single drug, but because there is a high number of drugs taken by at least 6 patients (18 according to Supplementary Table 3), I am not convinced that the overall the effect of treatment is negligible. Although I'm not an expert of treatments for ACVD, I assume that some patients are taking more than one drug, and some drugs can actually be similar in terms effects. Sorry if I missed some details, but can the author test the effect of taking any drug? This is possible only if there are patients that are treatment-naive. Otherwise they should at least try to test the changes in the microbiome associated with pairs of drugs rather than single drugs. In addition, they should include drug information as potential confounding factors in multivariate analysis.

2. I think that in general the analysis is well-performed. However, the final part on contrasting the microbiome in ACVD with other diseases can be a bit confusing for the reader as the paper deviates from the main point of the manuscript and does not add much to the discussion of the microbiome in ACVD. I suggest reducing drastically this part.

I also have a number of additional points and suggestions:

- Differently from similar studies (e.g. for a metabolic disease, type 2 diabetes, and others), in this study the analysis of the within-sample diversity (alpha diversity) is missing. The authors should perform this analysis (at the same sequencing depth) to check whether the ACVD microbiome is more or less complex than the microbiome in controls

- Looking at Figure 1, the much higher prevalence of Prevotella in controls is what drives most of the diversity between cases and controls. This is clear from Figure 1a, but is hidden in the boxplot of 1c as the Prevotella profile in controls is bimodal. A Fisher test on Prevotella high / Prevotella low association with disease / controls will likely result in a very significant p-value. Also, the difference in Prevotella prevalence is what drives the difference in the two networks of Figure 2, that should thus not be overinterpreted with other aspects.

- Figure 1: In the text is stated that PERMANOVA analysis was performed to test differences between ACVD and control samples with resulting a p-value less than 0.001 but in the caption is reported a p-value equal to 0.001 calculated with the Wilcoxon rank-sum test. Also, in both PCA and dbRDA analysis is not clear how the ordination plots were annotated with genus names (k-means?).

- Please expand the method section on dbRDA as this is not an analysis frequently performed in microbiome studies. Also, please define what CAP1 and CAP2 in Figure 1C mean and how they are computed

- Line 100: Please state p-values when say that there is a statistically relevant difference in abundance. Also, color-based legend for p-values in some figures are a bit confusing, but I'm not sure if this can be improved.

- Line 173: How the RF classifier would perform when comparing controls vs usage of combinations of drugs?

- In the methods paragraph describing the alignment of virulence factors using BlastP and the identification of bacteriophages with BlastN, the identity values used for the mapping are in my opinion and experience too low: this can lead to misclassification of bacteriophages and virulence factors. The presence of several bacteriophages that are usually species- or at least genus-specific without however identifying in the sample the corresponding host (e.g. the presence of a Mycobacterium phage without the presence of any Mycobacteria) are confirming this. Still based on my experience, several phages in the database are very closely related and it is likely that several reads map against more than one phage. Several phages detected as present are thus likely to be the same phage. The authors should perform clustering of phages used as reference to detect multiple variants of the same phage. Lastly, it is unclear whether the identified phages are indeed phages or are integrated prophages. The authors should at least discuss this in the manuscript.

- Line 238-239: this sentence is not very meaningful, please rephrase using statistical support.

- The manuscript uses both the terms "fecal microbiota" and "gut microbiota". Please consistently use only one of them. Also, sometimes the manuscript is still hard to follow for language problems it should be further checked for typos.

- Figure 3a: the boxplots are not useful as part of a main picture and they may be interpreted as real experimental values rather than predictions. The same information coming from the boxplot can be extrapolated from the ROC.

- Data deposition is very important for this kind of study as the dataset will be likely reused quite a lot. It is fair that the authors provide the public accession numbers only upon publication, but it would also be important to provide a table in the supplement linking all the available metadata (including clinical ones) with the future sample IDs as deposited in EBI.

Reviewers' comments:

Reviewer #2 (Remarks to the Author):

The authors have made an earnest effort to address the criticisms of all the reviewers and while there are still significant weaknesses, including low power and lack of causal inference, the paper also has important strengths:

1. This is the first metagenome wide association study (MWAS) for ACVD and can be used as a reference for future investigations on the role of microbiome composition in ACVD.
2. The number of individuals in the study is similar or greater than that of previously published MWAS in human disease cohorts.
3. Abundance of specific bacterial species and alterations in functional potential of the metagenome were correlated with ACVD.
4. Authors tested for the potential interaction of drugs on gut microbiome composition.
5. The authors identified novel commonalities and differences between the ACVD metagenome and metagenomes of other diseases.

We thank the Reviewer for these comments on the revised manuscript.

There are some details missing regarding sequencing depth and coverage. In the methods, the authors write that each sample was sequenced with “around 30 million paired-end reads.” The mean number of reads per sample and range should be disclosed in the results section. The sequencing coverage across the metagenome is not discussed, but could have important implications on the validity of results.

We apologize for this lack of clarity, and thank the Reviewer for this important point which is in fact a strength of our study. Sequencing statistics for each sample is shown in Sup. Table 1b. Mean sequencing was amounted to 56.5 million for raw reads and 55.2 million for high-quality non-human reads. We have now noted the numbers in the main text:

”After removal of low-quality and human DNA reads, 2.2 Tb of high-quality sequencing reads (55.2 million reads per sample) were aligned to a comprehensive reference gut microbiome gene catalog comprising 9.9 million genes¹⁹, which allowed on average 80.0 ± 3.5 % of the reads in each sample to be mapped (**Supplementary Table 1b**), consistent with saturation of the gene-coding regions^{4,19}.”

We have also double-checked the old Methods sections, which now also cite this Sup Table.

Reviewer #3 (Remarks to the Author):

The authors have attempted to address the numerous issues by removing data (metabolomics) and expanding sample numbers, or exploring microbiota taxa associations with alternative cardiometabolic conditions and trying to demonstrate unique associations between those identified in their expanded cohort for ACVD, vs alternative comorbidities.

While it is appreciated that the authors have attempted to address concerns, they did so by

providing virtually identical sorts of associative data, and fail to address the fundamental challenges and limitations of a purely associative exercise, which this study still remains. Like GWAS, MWAS identifies associations for further analyses; the 'guilt-by-association' is not to be overlooked, and would eventually lead to mechanistic insight. We believe that previous single disease-based MWAS studies have provided valuable results that are now subject for testing in the clinic. As was summarized in our recent review (Wang and Jia, 2016, Nat Rev Microbiol.), even for diseases such as obesity, for which causality has previously been shown with transplant or gavaging into germfree models, the strain- and gene-level associations made possible by MWAS could pinpoint new hypotheses for further mechanistic investigations and population screening. On the same track, we believe that one next step to take is to perform across-disease based association studies, as the one provided here for metabolic-related disorders. We are confident that such studies may have validity due to the overarching goal of identifying disease inter-linked microbiota changes that may be features of the diseases in question or predisposing to it, even if causality cannot be claimed from a single study.

The data remain associative and likely much will not replicate with an independent prospective cohort. As was recently shown for metagenomics analyses revealing associations with T2DM, associations previously reported as being signatures for T2DM that were even "validated" by others, ended up being associated with medication use (Metformin).

Thus, even if it associations reported in metagenomics data were to replicate, with the analyses performed, there are too many comorbidities and medications that are not adequately adjusted for. The "controls" are essentially healthy. - not propensity match (age, sex, comorbidities matched) subjects who are followed prospectively to show they do vs do not experience development of ACVD or adverse events.

We strongly caution against over interpretation of the study from Forslund et al. As is summarized in our recent review and confirmed in a very recent interventional study (Wu et al., 2017, Nat Med.), Forslund et al. made a clear case for the higher relative abundance of *E. coli* in metformin-treated vs. untreated individuals, which was also noted in the Supplementary Information of the second T2D MWAS from Karlsson et al.; the lower level of the recently named genus of *Intestinibacter* was also interesting. Despite the lack of adjustment for cohorts from different countries, the authors still identified higher *Clostridium bolteae* in untreated T2D vs. controls, lower butyrate-producing bacteria such as *Roseburia* spp. and *Clostridiales* spp. in untreated T2D vs. controls. They furthermore suggested that butyrate production was higher in metformin-treated vs. untreated individuals.

Larger validation cohorts would be possible with the continued drop in sequencing cost. We are looking forward to sequence and analyze prospective cohorts. Perhaps because metagenomics is still a young field, and collecting feces from the elderly requires much more effort than drawing blood samples, we have not been able to gather or collaborate with one such great cohort with high-quality sample collection and metadata recording.

While 5-fold folding sorts of analyses help to try and identify robust and reproducible associations of clusters of phenotypes, the approaches taken still have inherent limitations including:

1) being fundamentally associative only in nature, and failing to discriminate with other comorbidities.

2) Lacking in use of independent non-overlapping validation prospective cohort

3) Lacking in ability to identify microbiota, pathways or metabolites that are mechanistically or causally associated with the disease process.

Please refer to our responses above regarding the value of MWAS studies and the unavailability of a prospective cohort. We fully agree that it would be highly valuable if one was able to perform shotgun sequencing on fecal samples which could be linked with suggested functional analyses - and then further match it with a replication cohort. However, to our knowledge it will take some years from now before the field will reach to that point of resolution.

4) Moreover, the removal of the metabolomics data deteriorates the validity of the inferred Kegg pathway "functional" analyses. As the authors acknowledge, these are "hypothesis generating" exercises and lack validation. At least performance of validated stable isotope dilution LC/MS/MS analyses for candidate significant metabolites would have allowed for independent validation of the inferred functional association. So the removal of the data, rather than performance of the experimental studies and analyses, further weakened the manuscript, not improved it.

We very much appreciated the Reviewer's advice on metabolomics analyses in the [redacted] version and will continue to work on that end. While we also like the idea of demonstrating a microbiome-metabolome axis in ACVD, in practice metabolomics is still not nearly as high-throughput, and requires a handful of different platforms and protocols for a comprehensive investigation. We would like to note that identification of the metabolite TMAO alone took a dozen of high-profile publications, and in our opinion we still only know part of the story. After careful revision of the data, we decided to remove the metabolomics part, since it will need more validation to have full confidence in the data.

5) Kegg pathway analyses to infer function remains inference. Assigned enzyme activities are based solely on prediction based on sequence homology. Until actual biochemical data are experimentally tested for and verified, it is hypothesis only.

We are not entirely sure which enzymes the Reviewer are referring to, but do not think it would be fair for us to experimentally validate an entire database. Nonetheless, we now emphasize in the Discussion that experimental measurements of enzyme activities would be important (Page 13, Line 8).

Reviewer #4 (Remarks to the Author):

This study describes a metagenomic investigation of the microbiome associated with atherosclerotic cardiovascular disease (ACVD) based on 408 fecal samples collected from

Chinese individuals and metagenomically sequenced exclusively for this study. The dataset is then integrated with 845 samples from previously published studies. The sample size is in line with other metagenomic studies on the association between the microbiome and other diseases. Results showed the depletion of Bacteroides and Prevotella in ACVD patients, and enrichment of Escherichia and Streptococcus. Also Enterobacter aerogenes and Klebsiella pneumoniae are enriched in disease and these bacteria possess genes for TMA lysase whose activity is linked to atherosclerosis. Random forest classifier identified bacterial species that can be treated as features to discriminate ACVD samples from controls subjects. From a functional potential point of view, the ACVD microbiome share some similarity to the obesity and T2D one with an increase of PTS pathways and a decrease in the vitamin biosynthesis pathways (THF).

Overall, I think this is an interesting study which provides relevant results and data. However, I have two major issues to point out. The first is the potential problem of having treatment as confounder for disease. The second is the overwhelming amount and types of analyses that have been performed which are not always focused toward the main message.

We thank the Reviewer for interest in this first large cohort for ACVD and for the fresh comments. We provide detailed responses below.

1. Drugs as confounding factor. The authors try to address this potential problem in their revision. They show in the section “Influence of drugs on the gut microbiota” that drugs do have an influence on the gut microbiome but they somehow conclude that this effect is smaller than that of the disease. I can partially agree with their conclusion when considering each single drug, but because there is a high number of drugs taken by at least 6 patients (18 according to Supplementary Table 3), I am not convinced that the overall the effect of treatment is negligible. Although I’m not an expert of treatments for ACVD, I assume that some patients are taking more than one drug, and some drugs can actually be similar in terms effects. Sorry if I missed some details, but can the author test the effect of taking any drug? This is possible only if there are patients that are treatment-naïve. Otherwise they should at least try to test the changes in the microbiome associated with pairs of drugs rather than single drugs. In addition, they should include drug information as potential confounding factors in multivariate analysis.

We apologize for the ambiguity and have now listed the categories of medication in Sup Table 3, which shows no clear pattern separation at the category level. In addition to the single-drug analyses, we have now analyzed major drug combinations, which showed a very similar non-discernible pattern (new Sup. Table 4).

Furthermore, with and without ACVD remained the most significant factor in this cohort (p-value < 10^{-6} in PERMANOVA), independent of adjustment for medication (Supplementary Table 1c).

The main text has been revised accordingly to report on these new analyses (Page 5, Line 7; Page 8, Line 18).

2. I think that in general the analysis is well-performed. However, the final part on contrasting the microbiome in ACVD with other diseases can be a bit confusing for the reader as the

paper deviates from the main point of the manuscript and does not add much to the discussion of the microbiome in ACVD. I suggest reducing drastically this part.

We thank the Reviewer for the general assessment of our methodology. We however, find this a rare opportunity to compare ACVD with other cardiometabolic diseases, given their epidemiological and mechanistic overlaps, and the long history of studying T2D and obesity in the metagenomics field in contrast to ACVD.

I also have a number of additional points and suggestions:

- Differently from similar studies (e.g. for a metabolic disease, type 2 diabetes, and others), in this study the analysis of the within-sample diversity (alpha diversity) is missing. The authors should perform this analysis (at the same sequencing depth) to check whether the ACVD microbiome is more or less complex than the microbiome in controls

We have now analyzed both richness and diversity in the ACVD cohort and found no significant difference. The results are included in a .new Sup. Fig. 1.

- Looking at Figure 1, the much higher prevalence of *Prevotella* in controls is what drives most of the diversity between cases and controls. This is clear from Figure 1a, but is hidden in the boxplot of 1c as the *Prevotella* profile in controls is bimodal. A Fisher test on *Prevotella* high / *Prevotella* low association with disease / controls will likely result in a very significant p-value. Also, the difference in *Prevotella* prevalence is what drives the difference in the two networks of Figure 2, that should thus not be overinterpreted with other aspects.

While the previous version did not invoke the controversial concept of enterotypes, we think it would be interesting to look at our data in this light. *Prevotella* is probably more common in Chinese than in Europeans, although not to the same extent as samples from more undeveloped regions (Malawi, Burkina faso, Hazda). Consistent with previous 16S results (e.g. Arumugam et al. 2011, Nature; Falony et al. 2016, Science), *Prevotella* showed up in the PCA plot (Fig. 1a), but was not a top ranked ACVD-distinguishing genus in dbRDA (Fig. 1c). Both the *Bacteroides* and the *Prevotella* genera were higher in the controls (Fig. 1b); at the strain (MLG) level, there were a number of *Bacteroides* spp. while *P. copri* dominated the *Prevotella* genera (Fig. 2); the *Bacteroides* spp. and *P. copri* indeed showed negative associations (Fig. 2a), as would be expected for distinct enterotypes. To further illustrate this pattern, we show the assignment into enterotypes below, for the control and ACVD groups. The green box, i.e. the overgrowth of *Ruminococcus*, *Streptococcus*, *Klebsiella* etc. in ACVD (as in Fig. 2) was more notable than the decrease in *Bacteroides* or *Prevotella*.

- Figure 1: In the text is stated that PERMANOVA analysis was performed to test differences between ACVD and control samples with resulting a p-value less than 0.001 but in the caption is reported a p-value equal to 0.001 calculated with the Wilcoxon rank-sum test. Also, in both PCA and dbRDA analysis is not clear how the ordination plots were annotated with genus names (k-means?).

We apologize for this ambiguity. The original PERMANOVA analysis was performed with 999 permutations so the p-value was less than 0.001. We have now performed 99999 permutations, and the Results now reads:

” The ACVD and control samples were significantly different in multivariate analyses. ACVD status showed a p-value < 10⁻⁶ in permutational multivariate analysis of variance (PERMANOVA), whether or not medication was adjusted for (**Supplementary Table 1c**). The ACVD and control samples also showed separation in PCA (principal component analysis) and dbRDA (distance-based redundancy analysis) plots (**Fig. 1**)...”

No statistical test is needed for the PCA plot in Fig. 1a. As seen in other studies (e.g. <http://www.nature.com/nbt/journal/v26/n3/full/nbt0308-303.html>), for both PCA and dbRDA, genera where the largest weights in the principal component or coordinate are shown. The figure caption has been modified accordingly.- Please expand the method section on dbRDA

as this is not an analysis frequently performed in microbiome studies. Also, please define what CAP1 and CAP2 in Figure 1C mean and how they are computed
dbRDA is a supervised ordination technique designed to handle ecologically meaningful but non-Euclidean measures of dissimilarity. The dbRDA analysis was also used by studies such as Forslund et al. (Ref. 6) and Dethlefsen et al. (*PNAS*, 2010). We have now elaborated a bit more on this method both in the Figure caption and in the Methods (Page 24, Line 10; Page 15, Line 4).

- Line 100: Please state p-values when say that there is a statistically relevant difference in abundance. Also, color-based legend for p-values in some figures are a bit confusing, but I'm not sure if this can be improved.

All MLGs in Fig. 2 had a q-value < 0.05 in Wilcoxon-rank sum test controlled for multiple testing using the Benjamin-Hochberg procedure. This is now briefly noted whenever Fig. 2 is cited. The exact values are all listed in the accompanying Sup. Table 2.

- Line 173: How the RF classifier would perform when comparing controls vs usage of combinations of drugs?

Please refer to our above response regarding medication use. The RF results have been added as a new Sup. Table 4.

- In the methods paragraph describing the alignment of virulence factors using BlastP and the identification of bacteriophages with BlastN, the identity values used for the mapping are in my opinion and experience too low: this can lead to misclassification of bacteriophages and virulence factors. The presence of several bacteriophages that are usually species- or at least genus-specific without however identifying in the sample the corresponding host (e.g. the presence of a Mycobacterium phage without the presence of any Mycobacteria) are confirming this. Still based on my experience, several phages in the database are very closely related and it is likely that several reads map against more than one phage. Several phages detected as present are thus likely to be the same phage. The authors should perform clustering of phages used as reference to detect multiple variants of the same phage. Lastly, it is unclear whether the identified phages are indeed phages or are integrated prophages. The authors should at least discuss this in the manuscript.

We thank the Reviewer for the expert comments on phage analyses. We have now performed Spearman's correlation between the relative abundances of the phages and the MLGs. Judging from the positive correlations between the phages and their host bacteria (new Supplementary Fig. 8), we expect most of them to be integrated rather than free viral genomes, while formal detection of the lytic populations would require isolation of phage particles. We also agree with the Reviewer that with the limited knowledge of bacteriophages and their host range in the gut microbiome, it is difficult to uniquely distinguish taxonomically related phages, or establish a widely accepted cutoff. Due to these reasons, all the phage results are now in Supplementary Figures. The main text has been updated accordingly (Page 11, Line 11).

- Line 238-239: this sentence is not very meaningful, please rephrase using statistical support. We apologize for this ambiguity. In the original Fig. 6 (now Sup. Fig. 6), we showed phages that were significantly different in abundances between cases and controls in at least one of

the disease cohorts. +, q-value < 0.05; *, q-value < 0.01, Wilcoxon rank-sum test, FDR controlled.

- The manuscript uses both the terms “fecal microbiota” and “gut microbiota”. Please consistently use only one of them. Also, sometimes the manuscript is still hard to follow for language problems it should be further checked for typos.

We apologize for this inconsistency and are now using ‘gut microbiome’ throughout the manuscript. Feces are the most commonly used type of samples, and we consider it sufficient to mention it only once at the beginning and in the relevant Methods session.

- Figure 3a: the boxplots are not useful as part of a main picture and they may be interpreted as real experimental values rather than predictions. The same information coming from the boxplot can be extrapolated from the ROC.

We have now removed this redundancy and only show the ROC. The probability results are listed in Supplementary Table 6.

- Data deposition is very important for this kind of study as the dataset will be likely reused quite a lot. It is fair that the authors provide the public accession numbers only upon publication, but it would also be important to provide a table in the supplement linking all the available metadata (including clinical ones) with the future sample IDs as deposited in EBI.

We have added a column in Sup. Table 1 for the intended EBI sample IDs, as recommended by the Reviewer. The Accession code section now also cites this table.

REVIEWERS' COMMENTS:

Reviewer #4 (Remarks to the Author):

The authors have addressed most of my comments. I would just reiterate that highlighting the potential issue of the treatment as confounder (e.g. abstract) is in the interest of the authors.